# Weight-normative messaging predominates on TikTok—A qualitative content analysis

Marisa Minadeo☯, Lizzy Pope *☯

Department of Nutrition and Food Sciences, University of Vermont, Burlington, Vermont, United States of America

☯ These authors contributed equally to this work.
* efpope@uvm.edu

## Abstract

Tiktok is a social media platform with many adolescent and young adult users. Food, nutrition, and weight-related posts are popular on TikTok, yet there is little understanding of the content of these posts, and whether nutrition-related content is presented by experts. The objective was to identify key themes in food, nutrition, and weight-related posts on TikTok. 1000 TikTok videos from 10 popular nutrition, food, and weight-related hashtags each with over 1 billion views were downloaded and analyzed using template analysis. The one-hundred most viewed videos were downloaded from each of the ten chosen hashtags. Two coders then coded each video for key themes. Key themes included the glorification of weight loss in many posts, the positioning of food to achieve health and thinness, and the lack of expert voices providing nutrition information. The majority of posts presented a weight-normative view of health, with less than 3% coded as weight-inclusive. Most posts were created by white, female adolescents and young adults. Nutrition-related content on TikTok is largely weight normative, and may contribute to disordered eating behaviors and body dissatisfaction in the young people that are TikTok's predominant users. Helping users discern credible nutrition information, and eliminate triggering content from their social media feeds may be strategies to address the weight-normative social media content that is so prevalent.

**Data Availability Statement:** Links to videos and screenshots from each hashtag used in the study can be found below. The data is shared by hashtag so it is organized logically. #bodypositivity: https://doi.org/10.6084/m9.figshare.21082015.v1 #diet:

## Introduction

Social media is incredibly popular with young adults [1], and may be an arena where young adults are exposed to content that perpetuates diet culture. Diet culture is a system of beliefs that worships thinness, promotes weight loss as a means of attaining higher status, demonizes certain ways of eating while encouraging others, and oppresses people who do not match up with the prescribed vision of "health," most frequently women, trans people, larger-bodied people, people of color, and people with disabilities [2]. Several previous studies examined the presence of diet-culture content on social media, finding that content perpetuating the thin ideal and weight normativity is prevalent across various hashtags, #thinspiration, #fitspiration, #cheatmeal, #weightloss, #quarantine15, and social media sites such as Instagram and Twitter [3–8].

https://doi.org/10.6084/m9.figshare.21081952.v1 #fatloss: https://doi.org/10.6084/m9.figshare.21081757.v1 #mealprep: https://doi.org/10.6084/m9.figshare.21081673.v1 #plussize: https://doi.org/10.6084/m9.figshare.21070903.v1 #weightloss: https://doi.org/10.6084/m9.figshare.21070846.v1 #weightlosscheck: https://doi.org/10.6084/m9.figshare.21070783.v1 #whatieatinaday: https://doi.org/10.6084/m9.figshare.21067525.v1 #weightlossjourney: https://doi.org/10.6084/m9.figshare.21067510.v1 #nutrition: https://doi.org/10.6084/m9.figshare.21067504.v1.

**Funding:** The authors received no specific funding for this work.

**Competing interests:** The authors have declared that no competing interests exist.

Weight normativity posits that health is only possible at a specific weight, weight and disease are linearly related, and one has a personal responsibility for meeting weight expectations [9]. Because weight is seen as integral to health, the weight-normative approach focuses on weight management and achieving a "normal" weight. The ubiquitous nature of diet culture follows logically from weight normativity as weight management is seen by both as essential to health. In contrast to weight normativity, the weight-inclusive view of health recognizes that bodies come in a variety of shapes and sizes, and believes that people in all body sizes can achieve health if given the opportunity to pursue health behaviors and access to non-stigmatizing health care [9]. Importantly, weight-inclusivity does not define weight control as a health behavior. As weight-inclusive approaches are associated with improved physical and mental health outcomes, weight-inclusive messaging may help foster health promoting behaviors [10–14].

The presentation of diet culture, weight normativity, and the thin ideal on social media is problematic, as research indicates that social media usage in adolescents and young adults is associated with disordered eating and negative body image [15–17]. Although previous research has examined the food and weight-related content on several social media platforms quite thoroughly, the video-sharing app, TikTok, is a new and popular social media platform commonly used by young adults, and little to no research has examined the content of posts on TikTok. Since the app went worldwide in 2018, it has been downloaded over two billion times globally [18]. The app is like Instagram or Twitter where you can follow and like posts from certain accounts, but it is unique in several ways. The biggest difference with this increasingly popular social media site is that the app consists only of short videos created by its users. The app is very user friendly, providing a wide array of tools for creators to utilize such as filters, special effects and sounds from popular songs, TV shows, their own voice or other popular TikToks. TikTok is also notorious for trends; a certain dance, sound, prompt or hashtag will go viral and other users will then create their own versions. With TikTok, users don't need to follow certain accounts, or even have their own account, to view posts tailored to them. The default page on the app is its "for you" page, with endless, algorithmically curated videos based on content that a user has interacted with or watched previously. Therefore, the TikTok algorithm literally tailors content "for you." If someone consistently engages with diet, weight loss, or food content those videos will continue to appear unless the user actively selects a window labeled "not interested".

The potential exposure to endless weight or food-related content becomes more concerning when considering TikTok's user demographics. Most TikTok users are in Gen-Z (people born in the mid-1990's to mid-2010's). In July 2020, TikTok reported that one-third of its 49 million daily users were at or below the age of 14 [19]. Although TikTok recently created censorship policies on eating disorder content, [20,21] it is possible that the app still contains a substantial amount of content that reinforces the thin ideal, weight normativity, and diet culture, and may have the same negative impacts on eating behavior and body image as previous social media sites [16]. Conversely, if TikTok posts portray more weight-inclusive or body-positive content they could potentially help improve body image and feelings of acceptance [22,23].

Many health professionals may not even know what TikTok is or how to use it, making it impossible for them to counteract any dangerous messaging young adults may be absorbing on the app. The purpose of this study was to identify themes present in popular food, weight, and body-related posts on TikTok. We hypothesized that food, weight, and body-related posts would be very popular on TikTok, the most evident themes would be weight normative, and young people would be the most frequent content creators. Determining the themes around food and weight on TikTok can help health professionals better understand what messages young adults are seeing around food and weight to then counteract the effects of inaccurate or weight-normative content.

**Table 1. Initial list of food/weight/body hashtags.**

| Body image hashtag | Number of views | Eating behavior hashtag | Number of views |
|---|---|---|---|
| #weightloss | 9.7B | #health | 9.0B |
| #plussize | 2.6B | #healthy | 5.0B |
| #weightlossjourney | 2.1B | #whatieatinaday | 3.2B |
| #bodypositivity | 2.0B | #diet | 3.0B |
| #fatloss | 1.6B | #mealprep | 2.1B |
| #weightlosscheck | 1.3B | #healthyfood | 1.2B |
| #ED | 641.3M | #nutrition | 1.1B |
| #edrecovery | 638.4M | #dieting | 221.7M |
| #skinny | 421.6M | #nutritiontips | 185.9M |
| #weightlosstips | 365.6M | #calories | 201.2M |
| #bodyshaming | 264.1M | #intuitiveeating | 111.2M |
| #bodyimage | 183.2M | #dietculture | 60.1M |
| #skinnny | 146.7M | #metabolism | 58.6M |
| #bodyconfident | 79.9M | #antidiet | 22.5M |
| #haes | 15.3M | #healthymealideas | 9.0M |

## Methods

### Video collection

Hashtags are used on social media to collect content related to a specific topic. On TikTok, users can add hashtags to the caption of their post with the hash sign (#) prefacing a word or phrase. When users then click or search certain hashtags or topics, they will be brought to a feed of videos with that hashtag. In this study, initially, a list of thirty body-image and eating-related hashtags was collected by searching TikTok for food, nutrition, weight and body image related content, and noting which hashtags were commonly used by creators and had the most views. The thirty hashtags were generated both by brainstorming a possible list of food, weight, and body image-related words based on professional expertise in the nutrition field, as well as using TikTok for multiple weeks noting what hashtags were commonly used on food, nutrition, and weight-related posts (see Table 1).

All thirty hashtags had at least 9 million views per hashtag, and the most popular had 9.7 billion views. From this list of thirty, we eliminated the "health" and "healthy" hashtags because we felt they were too broad for the specific focus of the study, as they could include content on not smoking, wearing sunscreen, getting sleep, or other health behaviors beyond the focus of this study. The top ten most viewed hashtags from the original list were then included in this study except for choosing #nutrition with 1.1 billion views versus #healthyfood with 1.2 billion views, as it seemed important to analyze the content of videos purporting to be about nutrition which could be weight inclusive or perpetuate diet culture (Table 2). Since the posts were collected in September 2020, #nutrition now has more views that #healthyfood.

**Table 2. Final selected hashtag list.**

| Hashtag | Views |
|---|---|
| #weightloss | 9.7B |
| #whatieatinaday | 3.2B |
| #diet | 3.0B |
| #plussize | 2.6B |
| #weightlossjourney | 2.1B |
| #mealprep | 2.1B |
| #bodypositivity | 2.0B |
| #fatloss | 1.6B |
| #weightlosscheck | 1.3B |
| #nutrition | 1.1B |

Each of the ten hashtags had more than 1.1 billion views per hashtag and we believed were divided between hashtags that were likely to present weight-normative content (#weightloss, #diet, #weightlossjourney, #fatloss, #weightlosscheck) or potentially contain weight-inclusive or weight-neutral content (#whatieatinaday, #plussize, #mealprep, #bodypositvity, #nutrition). Although the original list of thirty hashtags contained several such as #intuitiveeating and #haes that may have been most likely to contain weight-inclusive content, those hashtags had considerably fewer views, 111.2 million and 15.3 million respectively than any of the top 10 hashtags used in this study which each had at least 1 billion views (see Table 1). These more specific weight-inclusive hashtags were then not selected for analysis because they did not represent a large share of the food/weight/body posts viewed on TikTok, and would not give an accurate idea of what the "average" TikTok viewer would find on their for you page, as the for you page is most likely to display videos with many views. The first 100 videos under each of the ten selected hashtags as well as a screenshot of their first frame and caption were downloaded in September 2020, and the files were each labeled with the hashtag and record number. The videos were publicly available and downloadable, so our collection and analysis complied with necessary terms and conditions. TikToks in each hashtag are displayed from the most views to the least views, so downloading the first 100 videos in each hashtag meant that the most viewed videos in each hashtag were collected. Any TikToks that were not downloadable were not included in the study, as a small percentage of users do not allow their videos to be downloaded. Because all videos were publicly available, the study was approved as exempt by the University of Vermont Committee on Human Research in the Behavioral and Social Sciences, STUDY00001190. As all of the TikTok videos analyzed were publicly available, informed consent was waived by the ethics committee.

## Codebook development

After preliminary analysis of a subset of 100 TikToks (10 from each hashtag), a list of codes was developed using a template analysis approach that included codes that we expected to find in the data such as posts that were food related, depicted exercise, reflected someone's body image, represented weight normativity or weight inclusivity, or discussed weight loss, as well as codes reflecting themes that emerged from the subset of TikToks [24]. Demographic codes for age, gender presentation, race/ethnicity, and body size were also included in the codebook, as was a code noting any professional degrees of the user. After codebook creation, two coders watched and coded ten videos from each of the hashtags. Codes from each coder were compared to ensure consistency of code application. Coders then proceeded to code the remainder of the TikTok videos independently. Once all coding was completed individually, cross comparisons were run in Excel between the two coders' spreadsheets to identify any differences in coding. Videos with discrepancies were viewed again by two coders together and final codes were agreed upon. In total, four coders worked on the project with two coders coding each video. The study authors trained the two additional coders who were nutrition undergraduate students. When coding discrepancies were noted, the two study authors looked at each video and determined final coding. All codes used in the codebook except for demographic codes can be found in the linked supplementary material file.

## Data analysis

Demographics were coded for videos in which the creator was visible in the video. Following the methods used by Lucibello et al., demographic codes were determined based on the perceived age (teenager, young adult, millennial, middle aged adult or elderly person), gender presentation (male, female, trans male, trans female, non-binary) and race/ethnicity (white

presenting or non-white presenting) of the person in the video [4]. The body size of the user (average/medium frame, thinner than average frame, and larger than average frame) was also coded for, in line with methods from previous research [4,6]. Users with health professional degrees were coded if they explicitly stated their profession in the video or the comments.

TikTok content was analyzed using thematic analysis with quantification, which allowed us to identify, analyze, and report key themes from the qualitative TikTok data [25]. Code frequencies were tabulated using SPSS Software (IBM Corp., Chicago). The frequency data helped identify how often various codes and resulting themes appeared in each hashtag. The research group reviewed and refined key themes together, and identified TikToks that were emblematic of each key theme. The analytic plan was pre-specified, and the hypotheses were determined before data was collected.

## Results and discussion

### Young female creators predominate on TikTok

Our results indicate that the majority of TikTok content analyzed was created by young users of high school (11.1% of posts) and college ages (42.4% of posts) versus millennials (28.2%) and those in middle age (3.3%). Age could not be determined in 15.2% of posts. When considering previous literature on the negative influence of social media on young people's body image and eating behaviors, there is reason to be wary of the impact of the app on its young adult users [26–28]. Most videos (64.6%) were created by female presenting users, as opposed to male presenting users (30.6%). Young females who create and engage with weight or food-related content on TikTok are at risk of having internalized body image and disordered eating behaviors from other aspects of their lives [29] making exposure to weight, food, or body-related content particularly troublesome. When observing race and ethnicity, white presenting individuals were most represented (56.1%) followed by non-white presenting individuals (32.6%). Thirty-four percent of TikToks were created by users coded to be of an average/medium frame, versus only 16.6% of posts showing someone with a larger than average frame. Therefore, the most common creators engaging with the nutrition-related hashtags, even those explicitly related to weight loss were actually those who most conformed to the thin ideal. These results are similar to explorations of the fitspiration, cheatmeal, and weightloss hashtags on Instagram where researchers also found a lack of body diversity, with mostly thin, muscular individuals depicted [3,6,7].

### The glorification of weight loss and frequency of a weight-normative perspective

When looking at the data, a striking theme was the prevalence of weight-related content across all ten hashtags. Nearly 44% of all the videos coded in this study had content about weight loss, and 20.4% of all videos explicitly showed a person's weight transformation in the video. Many of these videos followed the same format and included similar hashtags, filters, and sounds, speaking to the nature of TikTok and the ability for trends to become popular. Even, the whatieatinaday hashtag and mealprep hashtags that we believed at the beginning of the study may be weight-neutral and portray a variety of eating styles and meal preparations were quite weight normative with users showing how they meal prepped for a certain diet, or what they ate in a day to lose weight. In fact, the whatieatinaday hashtag has become so weight normative and triggering that videos using it now carry a trigger warning for eating disorders including a link to the National Eating Disorder Association's help line because so many people were using the hashtag to show how little they ate in a day [21].

The glorification of weight loss across many videos, and reoccurring suggestion that if you just try hard enough you can lose weight too, undoubtedly elevate the key principles of weight normativity, and may reinforce to viewers the belief that weight is an important indicator of health status and overall self-worth [9,30]. This danger is increased by the substantial number of views that these hashtags are receiving. The weight loss hashtag alone had almost 10 billion views at the time the videos were collected, showing that billions of people are interested enough in losing weight to engage with the hashtag. The number of views the weight-loss focused hashtags received vastly outnumbered the number of views more explicitly weight-inclusive hashtags received (see Table 1). In our study, less than 3% of all videos were coded for weight inclusive messaging or content, suggesting that weight-inclusive messaging is not prevalent across some of the most viewed nutrition, food, and body-related hashtags on TikTok.

Among the videos that depicted weight loss transformation, common themes included exercise routines and diet plans, often with images of routine weigh-ins and clothing "down-sizes." Twenty-two percent of videos depicted physical activity. Many of the videos depicting physical activity were also coded for weight loss, indicating that physical activity was being portrayed not for its inherent benefits to physical and mental health [31], but as a means to achieve weight loss. In multiple cases, the creator mentioned finally becoming "happy" after losing the weight, and how their journey to "better themselves" was not done yet, exemplifying diet culture's message that a person's body size is indicative of their health and moral status.

As discussed earlier, TikTok is unique in that users can choose from a variety of sounds to add to their videos. Several sounds were found to occur frequently as part of the weight loss trends and included language that poses weight loss as paramount. Dialogue, sounding like a pep talk from a coach or a trainer, containing phrases such as "no excuses," "get up" and "if you want it bad enough, you'll do it," implies that deciding not to pursue weight loss or being unable to lose weight is a personal motivation failure. These videos may give viewers the idea that intense weight loss transformations are both attainable and something to strive for–not only for appearance purposes but also for physical and mental well-being as previous research has illustrated the strong influence that media reinforcing the thin ideal has on people's self image [32,33]. Although some may argue that viewing weight transformation videos is motivating, Jebeile et. al found that adolescents in larger bodies reported that viewing successful weight loss videos felt unmotivating and discouraging when weight loss was portrayed as very easy, as it often is on social media [3].

Perhaps portraying weight loss would be less harmful if long-term weight loss was generally achievable. However, as Tylka et. al (2014) discussed in their literature review of weight normativity and weight inclusivity, weight loss interventions almost always fail; only about 20% of individuals who participate in weight loss interventions maintain the weight loss after one year, and this percentage decreases by the second year [9]. The collection of videos glorifying weight loss on TikTok represent a moment in time, but do not show the longer-term effects of weight loss interventions, such as weight-cycling, or repeated dieting and weight loss attempts over many years [34]. All these outcomes can result in negative impacts to both mental and physical health [35–37]. Furthermore, each time someone questing to lose weight is unable to keep the weight off, the perception that they must then be lazy or lacking in willpower is reinforced [38,39].

A large percentage (21%) of total users did portray a positive body image, however very few posted videos lacking weight-normative undertones. Often creators were positive about their body image *because* they had lost weight, and rarely were depicting body positivity for a body that would not be deemed "acceptable" by diet culture. Similarly, while there were videos that mentioned or showed a person's weight gain, these videos were the minority of weight-related

content and reinforced diet culture beliefs. Weight-gain content tended to be masked with body positive hashtags and mentions of "self-love," but still suggested that weight gain is inherently negative. Voice-overs and comments in these videos excused their weight gain or reassured themselves that it was okay to have gained weight. For example, a common caption would be something like, "I gained 20 pounds, but I still love myself." Having to state that you still love yourself when your weight increases suggests exposure to weight bias and fat phobia [40]. Such weight-related stigmas lead to social issues such as devaluation, discrimination and rejection of individuals who are in fat bodies [40].

**Food as a means of pursuing wellness.** Thirty-eight percent of videos explicitly showed food (cooking, eating, getting take-out etc.), and 11.9% of videos featured active cooking. A major theme that emerged about the food content in the chosen hashtags was that food content seemed to be devoid of pleasure or social/cultural influences, and instead was perceived as a means of pursuing health or wellness. A significant portion (14%) of all videos mentioned a specific diet or dieting. Repeated examples of fad diets from this subset of TikTok videos were high-protein or low-calorie diets, liquid "cleanses," and intermittent fasting. These tended to be posed as ways to achieve a certain body "goal." A noticeable number of users also shared videos of themselves making weight loss or detox teas or drinks, to which they attributed their weight loss. Videos like this may be especially deceptive for a viewer because often the diet or recipe is paired with a thin, attractive person, leaving the impression that the drink played a role in attaining the idealized body type. Another theme among food content was instructional videos of users showing how to make "healthy" versions of "junk" foods. Assigning good or bad labels to food brings emotion and morality to eating. These emotions are internalized as we eat, and eating a food deemed "bad" by diet culture's standards may lead to negative perceptions of self after consumption [41]. Moralizing food can cause hyper-awareness about food choices, and foster beliefs that certain foods should be avoided because they will cause weight gain or poor health. This can lead to development of eating disorders such as Orthorexia Nervosa, an eating disorder defined as the obsession with "correct" eating and a fixation on foods' role in our physical health [41,42].

**Nutrition advice for weight loss provided by non-experts.** Of all videos coded under the hashtag "nutrition," 47% provided some sort of nutrition advice. These videos primarily offered advice about what foods to eat for different purposes, mostly for weight loss, as one quarter of videos also referenced weight loss in addition to providing nutrition advice. An example pattern would be users showing their weight transformation, paired with explaining "what they ate on their journey." This suggests again that the purpose of food is to manipulate body size rather than for social or cultural fulfillment. Another key finding was the lack of professional representation on TikTok. Given the high percentage of videos that provided nutrition advice to viewers, it is surprising how few came from a health professional. Of all the videos, 1.4% were created by registered dietitians, suggesting very little expert nutrition advice on the app. Users without professional knowledge are sharing nutrition tips that can be inaccurate, and often for the purposes of weight loss. These types of videos likely spread and encourage harmful dieting interventions to a vulnerable audience that may not have strong media literacy skills. Health professionals should recognize that their young adult clients may be gathering nutrition information on TikTok, and that much of it is not evidence-based.

**Working to counter inaccurate and weight-normative content on TikTok.** As seen in the current study, content displaying diet culture themes is often present on social media, a primary source of information for many young adults. In regard to health behaviors such as eating, exercise, or body image, young people are vulnerable to the influence of social media content, and are not always able to discern which posts offer evidence-based advice and which do not [26,27]. Exemplifying the impact of media on body image, Tiggemann and Miller

found that adolescent girls who reported more time spent on social media were also more likely to have high internalization of the thin ideal [28]. This is not surprising when content on social media commonly represents ideas that are rooted in diet culture and weight normativity as seen in the current study. Acknowledging the weight-normative content on social media is important if health practitioners would like to help young people develop healthy relationships with food and their bodies.

The results of this research raise questions about how the dissemination of weight-centric messaging can be countered. Can the influence of TikTok and other social media apps be used to spread more positive and accurate information about food, health and weight? In September 2020, TikTok released a blog post where the company pledged to focus on safeguarding their community from harmful content, and stated their goal to support a body-positive environment for their users [20]. While it is significant that the platform has acknowledged the danger of diet culture content trending on their app, the content used in this study was collected after the initial changes were made, indicating that the problem is still pressing; even with PSA's and resources, diet culture remains a viral topic. For example, recent trends targeted quarantine bodies, promoting fad diets, exercise routines, and diet supplements advertised to help lose the weight gained during lockdown as we approach a post-pandemic lifestyle [43,44].

Perhaps diet culture can be combatted by the spread of weight-inclusive content, such as Health at Every Size and Intuitive Eating focused content, which work to celebrate the diversity of body sizes, and reject weight as a symbol of health and morality. A wide range of literature has found these weight inclusive models to have many benefits, such as improvement of eating disorder behaviors, as well as associations with improved physical and mental health outcomes [10–14,45,46]. Some research has also found that exposure to body positive content on social media improves body image for young women [22,47]. Increasing awareness and presence of these paradigms on social media could help mitigate the negative effects of diet culture messages on young viewers.

The dearth of expert voices on TikTok is difficult to combat, because it is challenging for experts like Registered Dietitians to garner views by mastering the TikTok algorithm, which often demands that one's content is appealing to adolescents and young adults. Social media success can be cultivated by experts but takes concerted creative effort, time investment, and the right persona. Especially on TikTok, videos go viral in a way that is largely controlled by the proprietary algorithm that places videos on users' for you pages. Therefore, although certainly it would help if more registered dietitians were able to gain attention on TikTok, it may be more realistic to help adolescents and young adults learn how to discern expert advice from unqualified advice by working on their media evidence analysis skills. Building media literacy for young adults is important, as previous research has found that media literacy may help decrease body dissatisfaction and thin-ideal internalization [48]. It may also behoove practitioners to discuss what type of content young adults are seeing on TikTok and how they could begin to avoid the weight-normative content on the app if they would like to by blocking or unfollowing particular accounts. Helping young adults curate their social media feeds is one way to reduce exposure to diet culture messaging. Future research should focus on how health experts can best engage with youth on platforms like TikTok.

## Strengths and limitations

This study is the first to our knowledge to examine nutrition and body-image related content on TikTok. TikTok's popularity especially among young people makes it an important driver of cultural trends, and therefore it is imperative to understand the content present on the app. The study also included a large sample of highly viewed videos, indicating that the themes

identified are representative of the content millions of users are engaging with. Additionally, the study deliberately coded for weight inclusive content in an effort to better understand if weight inclusivity is widely represented on a social media site. Previous work on social media has not specifically examined the presence or absence of weight inclusivity.

It is important to note that all videos were coded based on subjective observations by the four female coders. This means that gathering results such as user demographics (age, race, gender, body size) was not meant to be definitive, but rather to gather a general understanding of the people who are creating this type of content based on the available information that could be observed from the video. While differences in coder opinions were accounted for by having each video coded by two different coders, and then running cross comparisons, there is no way of knowing the exact demographics of each user beyond merely our observations. Additionally, not all TikTok videos were downloadable. A small percentage were skipped over when collecting the 100 videos from each hashtag because of this, and were not included in the study. Some of the videos appeared in more than just one of the selected hashtags, so the total 1000 videos that were coded included several duplicates, possibly reducing variety in the data. However, since the first 100 downloadable videos of each hashtag were chosen, duplicates in this study exist because they were popular enough to appear in the top viewed videos of more than one of hashtags, and so their inclusion in the study still provides valuable insights. The videos also only capture one point in time, and trends across TikTok change frequently. Therefore, it is possible that although the hashtags analyzed remain quite popular on the app, specific trends depicted in the video may have changed over time. Finally, as the study only examined one social media platform, the results cannot be generalized across various other social media outlets.

## Conclusions

This analysis found that nearly all the 1,000 TikTok videos collected displayed content that was notably weight normative. Key themes included glorification of weight loss, the positioning of food to achieve health and thinness, and the lack of expert voices providing nutrition and health information. Perhaps the most problematic finding from this study is that young people are most frequently engaging and creating diet culture content. The many trends associated with weight loss omit lifestyle factors that play a role in weight and health, and leave viewers with the message that weight loss and thinness is achievable and desirable to all, potentially leading to unhealthy perceptions and behaviors surrounding food, weight and body image [49]. Knowing what type of weight, food, and nutrition-related content is prevalent on TikTok is important so health professionals can better understand what type of messaging young people are engaged with, and begin to formulate strategies to counter the negative impacts that may arise from frequently viewing weight-normative content.

## Supporting information

**S1 Table. Codebook definitions and examples.**
(DOCX)

## Acknowledgments

The authors would like to thank Ellie Blom and Kira Mincar for their help with data analysis, both Ellie and Kira have given permission to be acknowledged.

## Author Contributions

**Conceptualization:** Marisa Minadeo, Lizzy Pope.

**Data curation:** Marisa Minadeo, Lizzy Pope.

**Formal analysis:** Marisa Minadeo, Lizzy Pope.

**Investigation:** Marisa Minadeo, Lizzy Pope.

**Methodology:** Marisa Minadeo, Lizzy Pope.

**Project administration:** Marisa Minadeo, Lizzy Pope.

**Supervision:** Lizzy Pope.

**Validation:** Marisa Minadeo, Lizzy Pope.

**Writing – original draft:** Marisa Minadeo, Lizzy Pope.

**Writing – review & editing:** Marisa Minadeo, Lizzy Pope.

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
