## [Decision Letter · Decision Letter 0]

15 Aug 2022

PONE-D-22-11377Weight-Normative Messaging Predominates on TikTok – A Qualitative Content AnalysisPLOS ONE

Dear Dr. Pope,

Thank you for submitting your manuscript to PLOS ONE. After careful consideration, we feel that it has merit but does not fully meet PLOS ONE’s publication criteria as it currently stands. Therefore, we invite you to submit a revised version of the manuscript that addresses the points raised during the review process.

ACADEMIC EDITOR: This is a very relevant topic nowadays, addressed in a well-written paper. Still, there are some concerns with regard to the contribution of the paper to the scientific literature, rigor and depth of analysis, and extrapolation of authors' interpretations of their results. Although our decision is minor revisions, some of these concerns are considered major issues. Thus, these aspects should be well addressed by the authors. Pay close attention to the reviewer's comments. As finding reviewers for this paper was very difficult at this moment, I have also reviewed the paper. Find attached the paper with my comments/suggestions.

We look forward to receiving your revised manuscript.

Kind regards,

Eliana Carraça

Academic Editor

PLOS ONE

Journal Requirements:

2. In your Methods section, please include additional information about your dataset and ensure that you have included a statement specifying whether the collection and analysis method complied with the terms and conditions for the source of the data.

3. You indicated that ethical approval was not necessary for your study. We understand that the framework for ethical oversight requirements for studies of this type may differ depending on the setting and we would appreciate some further clarification regarding your research. Could you please provide further details on why your study is exempt from the need for approval and confirmation from your institutional review board or research ethics committee (e.g., in the form of a letter or email correspondence) that ethics review was not necessary for this study? Please include a copy of the correspondence as an "Other" file.

Additional Editor Comments:

This is a very relevant topic nowadays, addressed in a well-written paper. Still, there are some concerns with regard to the contribution of the paper to the scientific literature, rigor and depth of analysis, and extrapolation of authors' interpretations of their results. Although our decision is minor revisions, some of these concerns are considered major issues. Thus, these aspects should be well addressed by the authors. Pay close attention to the reviewer's comments.

Reviewers' comments:

Reviewer's Responses to Questions

**Comments to the Author**

1. Is the manuscript technically sound, and do the data support the conclusions?

Reviewer #1: Yes

2. Has the statistical analysis been performed appropriately and rigorously? 

Reviewer #1: N/A

3. Have the authors made all data underlying the findings in their manuscript fully available?

Reviewer #1: Yes

4. Is the manuscript presented in an intelligible fashion and written in standard English?

Reviewer #1: Yes

5. Review Comments to the Author

Reviewer #1: This is an interesting study presented well however there are some concerns with regard to the contribution to scientific literature and rigor of methods.

1. The analysis done is a content analysis with quantification - please clarify this in the methods

2. There are some findings presented in the discussion such as - line 27 on page 5 An example pattern would be users showing their weight transformation, paired with explaining “what they ate on their journey.” and line 245 on page 13 Dialogue, sounding like a pep talk from a coach or a trainer, containing phrases such

as “no excuses,” “get up” and “if you want it bad enough, you’ll do it,” implies that not trying to

lose weight makes you lazy, and inferior to those who are pursuing weight loss. These findings have not been presented in the results..the analysis could include a thematic analysis -all transcripts (text) could be coded and then themes identified.

3. Much of the discussion reads meaning into the text of the videos this is not scientifically appropriate nor robust as a method

4. Limitation - please mention that this reflects only one social media platform and the study would have been richer if other platforms such as Instagram would have been included

5.Please specify clearly what time period the data was collected- some places 'fall' is mentioned and another place September...this analysis is very sensitive to time and what was trending then may not be trending now..please mention in limitations

6. The discussion currently presents new data with interpretation..please consider deepening the analysis on what the study tells us, who it would benefit what kinds of action can be recommended ..how this study contributes or relates to other scientific literature.

6. PLOS authors have the option to publish the peer review history of their article (what does this mean?). If published, this will include your full peer review and any attached files.

Reviewer #1: No

---

## [Author Response · Author response to Decision Letter 0]

12 Sep 2022

Reviewer #1: This is an interesting study presented well however there are some concerns with regard to the contribution to scientific literature and rigor of methods.

1. The analysis done is a content analysis with quantification - please clarify this in the methods

We have clarified that the analysis was a thematic analysis with quantification.

2. There are some findings presented in the discussion such as - line 27 on page 5 An example pattern would be users showing their weight transformation, paired with explaining “what they ate on their journey.” and line 245 on page 13 Dialogue, sounding like a pep talk from a coach or a trainer, containing phrases such as “no excuses,” “get up” and “if you want it bad enough, you’ll do it,” implies that not trying to lose weight makes you lazy, and inferior to those who are pursuing weight loss. These findings have not been presented in the results..the analysis could include a thematic analysis -all transcripts (text) could be coded and then themes identified.

We did conduct a thematic analysis, where we coded all videos including their dialogue, sounds, and actions to identify predominate themes. We have restructured our paper with a combined results/discussion divided by theme to more clearly delineate the themes we identified in the data.

3. Much of the discussion reads meaning into the text of the videos this is not scientifically appropriate nor robust as a method

We respectfully disagree that the discussion reads meaning into the text of the videos. The discussion section identifies themes that emerged from the videos and then uses examples from the videos to illustrate those themes. The discussion also explains why these themes may be helpful or hurtful to a viewer. The authors are not implying that a particular creator meant to for example, glorify weight loss, but any video that spoke positively of weight loss was coded into this theme, and then previous research has indicated how the glorification of weight loss may be harmful. In the discussion we hoped to help readers realize why the themes we identified were important, we meant to provide meaning to our themes, not to interpret particular videos beyond categorizing them into our themes. We have made sure to tone down any causative language that may have been in the discussion which hopefully will help with this concern. If you can provide specific examples of where you feel we have over-reached in our discussion, we are happy to revise.

4. Limitation - please mention that this reflects only one social media platform and the study would have been richer if other platforms such as Instagram would have been included

We have added this as a limitation.

5.Please specify clearly what time period the data was collected- some places 'fall' is mentioned and another place September...this analysis is very sensitive to time and what was trending then may not be trending now..please mention in limitations

The posts were collected in September 2020, we have specified this throughout the manuscript. We have added a limitation about how trends continue to change on TikTok, although the hashtags we analyzed have substantially more views today than they did when we analyzed them, indicating that they continue to be popular on TikTok.

6. The discussion currently presents new data with interpretation..please consider deepening the analysis on what the study tells us, who it would benefit what kinds of action can be recommended ..how this study contributes or relates to other scientific literature.

We have restructured the paper to have a combined results/discussion section presented by theme. We feel that this structure better presents our quantitative and qualitative data. In the discussion section we have also added several tie ins to previous literature on social media. We have recommended that adults help young adults curate their social media feeds and improve their evidence analysis skills. We also think it’s important for experts to begin to engage on social media platforms to highlight weight-inclusive content. We have highlighted these suggestions in our discussion. As this paper didn’t study possible solutions to the weight-normative inaccurate content on TikTok, we can only offer suggestions for future research.

Editor’s Comments

We believe we have addressed all of the editor’s comments. Thank you.

---

## [Editor Report · Decision Letter 1]

26 Sep 2022

PONE-D-22-11377R1Weight-Normative Messaging Predominates on TikTok – A Qualitative Content AnalysisPLOS ONE

Dear Dr. Pope,

Thank you for submitting your manuscript to PLOS ONE. After careful consideration, we feel that it has merit but does not fully meet PLOS ONE’s publication criteria as it currently stands. Therefore, we invite you to submit a revised version of the manuscript that addresses the points raised during the review process.

ACADEMIC EDITOR:The authors have replied satisfactorily to all reviewer's queries. There are still some minor issues to address. Please make those amendments and resubmit and improved version of the paper. Please check the attached file.

We look forward to receiving your revised manuscript.

Kind regards,

Eliana Carraça

Academic Editor

PLOS ONE
---

## [Author Response · Author response to Decision Letter 1]

26 Sep 2022

Journal Requirements:

We have checked all references and do not believe any articles have been retracted. We also updated the reference formats for several of the references to make sure they had correct links included. We are happy to update additional references if anything has been missed. Thank you!

---

## [Editor Report · Decision Letter 2]

4 Oct 2022

PONE-D-22-11377R2Weight-Normative Messaging Predominates on TikTok – A Qualitative Content AnalysisPLOS ONE

Dear Dr. Pope,

Thank you for submitting your manuscript to PLOS ONE. After careful consideration, we feel that it has merit but does not fully meet PLOS ONE’s publication criteria as it currently stands. Therefore, we invite you to submit a revised version of the manuscript that addresses the points raised during the review process.

We look forward to receiving your revised manuscript.

Kind regards,

Eliana Carraça

Academic Editor

PLOS ONE

Journal Requirements:

Additional Editor Comments:

Dear Dr. Pope,

Please, resubmit your paper addressing all the editor's comments. Thank you.
---

## [Author Response · Author response to Decision Letter 2]

4 Oct 2022

PLOS ONE Response To Reviewers 9/26/22

Journal Requirements:

We have checked all references and do not believe any articles have been retracted. We also updated the reference formats for several of the references to make sure they had correct links included. We are happy to update additional references if anything has been missed. Thank you!

Editor’s Comments

Lines 243-246. Not sure you can make this interpretation. I believe you could say that those sentences imply that one might be lazy or lack strong will to move or do something, bu I cannot see how it can directly imply inferiority and social comparison. Please adjust sentence. Also, find some references that support this interpretation of yours.

We agree that perhaps the dialogue does not imply social comparison directly. We have revised and removed the social comparison language, replacing it with an interpretation that the dialogue implies that not being able to lose weight is a personal failing of motivation. We hope this edit addresses the editor’s concern.

Adjust sentence. There are two "reported". Sentence is confuse.

Thank you, the sentence has been revised.

---

## [Editor Report · Decision Letter 3]

20 Oct 2022

Weight-Normative Messaging Predominates on TikTok – A Qualitative Content Analysis

PONE-D-22-11377R3

Dear Dr. Pope,

We’re pleased to inform you that your manuscript has been judged scientifically suitable for publication and will be formally accepted for publication once it meets all outstanding technical requirements.

Kind regards,

Eliana Carraça

Academic Editor

PLOS ONE
---

## [Editor Report · Acceptance letter]

24 Oct 2022

PONE-D-22-11377R3 

Weight-normative messaging predominates on TikTok – a qualitative content analysis 

Dear Dr. Pope:

I'm pleased to inform you that your manuscript has been deemed suitable for publication in PLOS ONE. Congratulations! Your manuscript is now with our production department. 

Kind regards, 

on behalf of

Dr. Eliana Carraça 

Academic Editor

PLOS ONE